# Mechanisms Determining Body Size and Shape Difference in Algerian Spur-Thighed Tortoises (*Testudo graeca*)

**DOI:** 10.3390/ani12101330

**Published:** 2022-05-23

**Authors:** Manel Tiar-Saadi, Ghoulem Tiar, Zihad Bouslama, Pavel Široký

**Affiliations:** 1Functional and Evolutionary Ecology Laboratory, Faculty of Natural and Life Sciences, University Chadli Bendjedid, 76, El Tarf 36000, Algeria; bihorau@yahoo.fr (M.T.-S.); tiarghoulem@yahoo.fr (G.T.); 2Environment and Biodiversity Research Division, National Environmental Research Center, Sidi Amar University Campus, 2024, Annaba 23005, Algeria; 3National Environmental Research Center, Sidi Amar University Campus, 2024, Annaba 23005, Algeria; z.bouslama@mesrs.dz; 4Department of Biology and Wildlife Diseases, Faculty of Veterinary Hygiene and Ecology, University of Veterinary Sciences Brno, Palackého 1946/1, 612 42 Brno, Czech Republic; 5CEITEC-Central European Institute of Technology, University of Veterinary Sciences Brno, Palackého 1946/1, 612 42 Brno, Czech Republic

**Keywords:** *Testudo graeca*, sexual dimorphism, geographic variation, body size, shell shape, proximate causes

## Abstract

**Simple Summary:**

Assessing the body size and body shape variations between sexes and geographical populations can help us understand the adaptive responses of organisms in the face of the pressures to which they are subjected. To evaluate the influence of habitat-type conditions, we selected six Algerian populations of *Testudo graeca* living in different environments. The results of the traditional morphometric analyses showed that body size and shell shape were smaller and flattened, respectively, in males, especially under unfavorable conditions for tortoises; these changes were jointly caused by anthropogenic and natural pressures. We found clear evidence in several tortoise species that differences in growth durations up to the onset of maturity resulted primarily in different sizes at maturity and ultimately in different adult sizes.

**Abstract:**

Using data for the body size and shell shape of Algerian *Testudo graeca*, we assessed how proximate causes shaped the observed variation in the morphology of adults. All of the studied populations displayed significant sexual size and shape dimorphisms. Relative to body length, females displayed larger, more voluminous and domed shells than males. We found clear evidence that variation in body size at maturity influenced sexual size dimorphism. Body size at maturity depends on the duration of growth from hatching up to the point of reaching sexual maturity. In the studied populations, sexual maturity, estimated by counting growth lines, was always reached earlier in males than in females (a time difference of 1.4–3.0 years). Similar to sexual size dimorphism, geographic variation in adult body sizes was also influenced by variations in the corresponding sizes at maturity. Remarkably, the population with the largest tortoises had the latest mean maturation time: 9.1 for males and 10.5 for females. Thus, the later completion of maturation was a determinant for a larger size in adulthood. The largest tortoises among the studied populations were measured at the Djelfa locality, where the recorded sizes of males and females reached 186 and 230 mm, respectively.

## 1. Introduction

The spur-thighed tortoise *Testudo graeca* Linnaeus, 1758, is widely distributed throughout the Mediterranean and the Middle East as far as Easternmost Iran [1]. This species’ North African distribution extending from the Moroccan Atlantic coast to the Libyan Cyrenaica Peninsula includes five recognized lineages [2]. Their nomenclature has been changed recently to the five following subspecies: *T. g. graeca* (W-central Morocco), *T. g. marokkensis* (NW Morocco and central), *T. g. whitei* (NE Morocco and N Algeria), *T. g. nabeulensis* (far NE Algeria and N Tunisia) and *T. g. cyrenaica* (N Libya) [3,4,5]. While the phylogeographic structure of *T. graeca* in the Western Mediterranean area is well studied, data on the ecology and morphology of these populations remain incomplete and disproportionate by region. A number of studies carried out in Morocco, e.g., [6,7,8,9,10], Spain, e.g., [11,12], and Greece, e.g., [13,14], make *T. graeca* one of the best-studied tortoises in the Western Palearctic. In comparison, Algeria remains an understudied area offering little data for a single population living in a humid climate [15], currently assigned to *T. g. nabeulensis* [2], which cannot serve as a simple proxy for the predictive models of the population ecology of the *T. graeca* complex in otherwise prevailingly arid zones. The phylogeographic study by Fritz et al. [2] conducted mainly on Northern populations in Algeria revealed the dominance of the subspecies *T. g. whitei*, especially when moving away from the Tunisian borders in the extreme North. Algerian *T. graeca* colonized various habitats distributed throughout the Northern part of the country, between the coastlands in the North and in the limits of the desert in the South; from humid to hyper-arid habitats; and from sea level up to areas with altitudes of around 1400 m in the Saharan Atlas [16]. In the present study, we selected several populations in Southern Algeria, some of which are located more than 500 km away from the Southernmost population identified as *T. g. whitei* by Schweiger and Gemel [3]. The confirmation of at least two distinct subspecies in Algeria, and the uncertainty of other unverified populations located further South, motivated us to study several Algerian models and compare them with conspecific populations. 

A study based on a broad spectrum of chelonians concluded that variation in body size is mostly due to habitat differences [17]. Body size may also vary between conspecific males and females, forming sexual size dimorphism (SSD). Although the variation and quantification of the tortoise SSD have been commonly studied [8,13,18,19], their relationship to local environmental conditions is rarely measured in the literature. Previous studies assume that natural selection favours larger females with higher fecundity [13]. The pressure of sexual selection that affects the male body size results in optimizing the chances of courtship and copulation [18]. These divergences between the two prominent selection pressures—natural and sexual—are considered to form the basis for the differential sizes of bodies between males and females. Three main mechanisms may affect the intensity of SSD: differential sizes at hatching [20], differences in growth durations, and/or growth rates [6,14]. This phenomenon deserves to be verified in *T. graeca* populations, which consistently exhibit differences in body size, which could be used to explain the assumed size differences through different habitat-type conditions. 

Any characterization of tortoise morphology must consider shell shape. Although sexual dimorphism of the overall body shape, known as sexual shell shape dimorphism (SShD), has been previously studied in some chelonian species, only three studies to date have dealt with *T. graeca* [7,8,21]. Nevertheless, these studies did not verify the determinism of the environment on selective pressures or on the degree of dimorphism in distinct habitats. The assessment of morphometric variations between sex and different geographical populations could help us understand the adaptive responses of organisms under the pressures to which they are subject. If conspecific populations, subject to distinct local environmental conditions, differ remarkably in the direction or magnitude of their SSD and/or SShD, this difference will provide us with an opportunity to measure the actual extent of these factors on the morphometry of tortoises and to assess whether the same environmental constraints lead to the same ‘adaptive solutions’. 

To attain decisive conclusions, we have selected six Algerian populations of *T. graeca* living under different environmental conditions and various scales of anthropogenic pressure, located between the littoral area in the Northeast and the Saharan Atlas in the Southwest. We set the following objectives for the paper: (i) to ascertain whether body size and body shape vary with sex and geography, (ii) to verify the impact of the proximate cause ‘the duration of growth up to the onset of maturity’ in the observed differential morphology of adult *T. graeca*, and (iii) to evaluate the influence of habitat-type conditions on the morphology of the tortoise populations living in different habitats in Northern Algeria.

## 2. Materials and Methods

### 2.1. Study Areas 

Six sampling areas were selected to represent the variability of tortoise habitats in Algeria (Figure 1). Localities are distributed between the low-altitude Mediterranean coastline zone at 10–90 m a.s.l. (El. Kala, Annaba), through the Central Aures Mountains with steppe habitats of around 900 m a.s.l. (Batna), up to the Saharan Atlas covered by forests (Djelfa, 190 km^2^ of the last forest in the steppe belt before the onset of desert) or steppe habitats (Laghouat, Aflou) with altitudes of over 1200 m a.s.l. The characteristics of all localities are summarized in Table 1. 

### 2.2. Data Collection

Tortoises were collected by hand between March and early June within the period of 2006–2017 and individually marked on the marginal scutes of the carapace with a unique code devised by Stubbs et al. [22]. Sampling was carried out only on mature specimens, for which the sex, from external morphological criteria, was readily determined. The plastron area, supracaudal scute curvature, tail shape, and relative position of the cloaca were used as sexing criteria and can only be distinguished using a sufficiently large body size reached at maturity [6,7,21,23]. Hatchlings and juveniles were excluded from this study and were distinguished from adults by the appearance of reduced rings on their scutes as a consequence of reduced growth after reaching maturity (see more details below). We determined in the field that individuals with no narrow rings did not exhibit sexual behavior. Three main reproductive behaviors were examined: courtship, copulation, and positive palpation for the presence of eggs. 

For both characterizing the external morphology of Algerian *T. graeca* and comparison purposes, we selected the same main morphometric descriptors of body size and shell shape previously used in the literature, e.g., [7,8,13,14,17,18,19,21,24]. We measured eight main body size measurements as described and illustrated in Figure 2. Seven measurements concerned the three dimensions of the tortoise shell (length, width, and height) on the two dorsoventral sides (carapace and plastron), including straight carapace length (SCL), curved carapace length (CCL), mid-body carapace width (MCW), anterior carapace width (ACW), posterior carapace width (PCW), plastron length (PL), and shell height (SH). The eighth measure, anal notch width (ANW), concerned the width of the space available to lateral tail movements between the posterior tips of the anal scutes.

By analogy with studies carried out on tortoise shell shape models, we used the body proportion traits (size-corrected morphological traits, performed via Analyses of Covariance (ANCOVA) for comparisons between the sexes and regions, e.g., [7,8,13,18,19]. ANCOVA with the factor of sex or sex region and covariate SCL was used to test the differences in all body measurements except for SCL, for which only the covariate sex and sex region were considered, resulting in seven body shape traits analyzed in this study. The proportional difference between CCL and SCL may provide information on the domed shape of the shell. SH relative to SCL can provide information on the belly shape of the plastron. Body width descriptors (ACW, MCW, and PCW) relative to SCL were used to provide a proportional measure for the geometric shape of the shell in width to assess the pectoral, abdominal, and posterior width of the shell, respectively. PL relative to SCL indicates the dimensions of the opening available for movement of both the head and tail. ANW relative to SCL was used to estimate the space available for lateral movement of the tail. All linear measurements were performed using calipers (with a precision of 0.1 mm), except for CCL, which was gathered with a flexible rule (with a precision of 1 mm). 

Age and size at maturity

Successive growth rings deposited on the shell scutes corresponded to annual marks of growth and growth halt since hatching, relating to periods of activity and hibernation, respectively. The redirection of energy allocation from growth to reproduction just after the achievement of sexual maturity resulted in a considerable decrease in the growth-ring width, which became narrower [25,26]. Such tight rings formed after maturation were recognizable but hard to count with precision. These rings were not counted, and only the age at maturity was estimated as the number of broad growth rings. Counting was carried out on the right second costal scute and was systematically corroborated using the abdominal and pectoral scutes. This method is based on the factor that one complete ring corresponds to a cycle of one year (a 1:1 ratio) and is the most extensively used approach for age estimate in turtles and tortoises, including *T. graeca* [6,7,12,15,25,26]. Compared to growth models using Bayesian inference [27] and skelotochronology [28], the growth ring counting method is easier to perform and harmless to the studied animals. Wilson et al. [29] reported that the accuracy of the technique is reliable for only some chelonian species, for which the deposition of rings follows the assumed rate of one ring per year. Rodríguez-Caro et al. [30] confirmed this method as reliable for *T. graeca* age estimation, especially for young specimens, and deducing sexual maturity. We used this method only for the determination of sexual maturity.

The mean body size at maturation (SCLα) of both males and females was calculated among individuals that had just reached sexual maturity, which was distinguished by the appearance of the first tight ring on the scute [13,21]. After analysis, all sampled specimens were released at the site of collection. 

### 2.3. Climatic Data

Climatic data were collected for each locality from the nearest meteorological station and referred to as the average values over the last 20 years. These data were transformed into climatic parameters and indexes and tested for their correlation with the morphological variations of the tortoises. Four thermal parameters were selected, including the mean daily maximum and minimum temperatures for the year (the most general measures of the thermal climate) and the mean daily maximum and minimum temperatures for March, when tortoises begin to be active. Temperatures in March are likely to reflect the length of the activity season [14]. The effective temperature index (Eff-T) provides a measure of both the temperature and the length of the warm season and correlates well with the distribution of reptiles [14]. Eff-T was calculated following Stuckenberg [31] in Equation (1):(1)$$\mathrm{Eff}-T\;=\frac{8Ť+14\mathrm{Tr}}{\mathrm{Tr}+8}$$
where Ť is the mean annual temperature, and Tr is the annual range of temperature. The mean temperature in each month was calculated as follows: (mean daily max + mean daily min)/2, where Ť is the mean temperature across all months, and Tr is the difference between the means of the warmest and coldest months. The precipitation concentration index (PCI; ref. [32]) is an indicator of the received amount of rainfall, where higher values indicate more concentrated precipitation in a specific period and can define the bioclimatic stage. PCI was calculated at both annual (the sum of the rainfall received throughout the year in millimeters) and supra-seasonal scales for wet (October to March) and dry (April to September) seasons as (2) annual and (3) supra-seasonal: (2)$$\mathrm{PCIannual}=\frac{\sum_{i=1}^{12}pi^{2}}{{\left(\sum_{i=1}^{12}pi\right)}^{2}}\;100\;$$
(3)$$\mathrm{PCIsupra}\;\mathrm{seasonal}\;=\frac{\sum_{i=1}^{6}pi^{2}}{{\left(\sum_{i=1}^{6}pi\right)}^{2}}\;50$$
where *p_i_* is the monthly precipitation in millimeters for month *i*. An annual aridity index, De Martonne [33] (A_dM_), which assesses the land degradation resulting from climatic variations (low values indicate a prolonged drought due to low rainfall and high summer temperatures such that the vegetation has little opportunity to be restored), was calculated as follows in Equation (4): (4)$$A_{\mathrm{dM}}\;=\;\frac{P}{Ť\;+\;10}$$
where P is mean annual rainfall in millimeters.

### 2.4. Statistical Analysis

Statistical analyses were performed using Minitab Version 16.0 (Minitab, Inc., State College, PA, USA, https://www.minitab.com/en-us/), accessed on 1 March 2020. All variables were tested for normality and homogeneity of distribution using the Kolmogorov–Smirnov and Levene’s tests, respectively, before analysis of variance (ANOVA, MANOVA) or covariance (ANCOVA, MANCOVA). Variables characterizing body size were taken as the means, and variables for body shape were the adjusted means (body proportions), with SCL as the covariate. Multiple analysis of variance (MANOVA) was performed to compare overall tortoise body sizes between sexes, localities and the interaction between these two factors. In the same way, multiple analysis of covariance (MANCOVA) was performed to compare overall body shapes, using SCL as a covariate and the other variables as responses. Analysis of variance (ANOVA) and of covariance (ANCOVA) was used to assess both sexual and geographic variations in body size and the shell shapes of *T. graeca* specimens, respectively. Morphometry statistics used a one-way interaction by sex and two-way interaction by sex × site. Geography statistics used one-way interactions for males and females separately. We also applied Tukey’s post hoc test to explore differences between the averages of body size measured in the studied populations and to determine which group averages are significantly different from others. This test compares all possible pairs of means. Geographical characteristics (latitude, longitude, and altitude) were tested for their correlation with the morphological variations of the tortoises. In all statistical tests, a *p*-value of <0.05 was considered a significant difference. The percentages of both body size SSD% and shell shape sexual dimorphism SShD% were calculated at both maturity and during adulthood according to Willemsen and Hailey [13], expressed as follows: (100 (Female − Male)/Male). Growth during adulthood was deduced in each population by subtracting the mean SCL of adult individuals and their corresponding mean SCLα (mean body length among the individuals in the first year of maturity). We also quantified the mean relative difference between the two sexes in growth gain (in millimeters) after the acquisition of maturity (adult phase) according to the following formula: (Female − Male)/Male). Both degrees of geographic variation in size (GVS) and in shell shape (GVSSh), using all characteristics, were calculated between males and also between females as: 100 ((Population of largest specimens − population of smallest specimens)/population of smallest specimens). To analyze demographic mechanisms (proximal)-affecting adult body sizes, SCL was preferred as an indicator of body size. Sexual and geographic differences in body size may have several proximate causes. In the present study, we evaluated the relationships between size acquired at both maturity and during adulthood, age at maturity, adult body size, SSD, and geographic variation in body size (GVS). A principal components analysis (PCA) was also performed to select the largest contributor between the climatic variables to the variation in body size and shape and age at maturity.

## 3. Results

### 3.1. Sexual Size and Shell Shape Dimorphism

#### 3.1.1. Sexual Size Dimorphism (SSD)

Multivariate comparison using the absolute values of the set of eight morphometric size measurements revealed an overall variation between the two sexes without reference to their geography (MANOVA, sex Wilk’s λ = 0.34; *F*_8, 369_ = 89.89; *p* < 0.001). Based on the one-way ANOVA, a significant difference in each of the eight size descriptors was recorded between the two sexes independently of geographic origin (Table 2), thus confirming the SSD of body length, width, and height in all studied Algerian tortoise populations. The females in each population had significantly longer measurements than males for both the carapace (SCL and CCL) and plastron (PL). Females also had significantly wider bodies on all three measured lines (ACW, MCW, and PCW) and higher shells (SH) (Table A1 in Appendix A, Figure 3). Conversely, the anal notch width (ANW) was significantly greater in males, providing them with greater space for lateral movement of the tail and facilitating successful copulation.

A significant difference in body size between sexes among the six populations was also recorded via MANOVA using all eight morphometric size measurements (MANOVA, sex × site Wilk’s λ = 0.07; F_88, 2363_ = 13.04; *p* < 0.001). The two-way interaction test (sex × site) of each of the measured characteristics was always significant between the six studied populations (Table 2), and the degrees of SSD in the measured general descriptors varied among localities (range, 13–26%). The minimum SCL dimorphism was observed in Annaba, where females were, on average, only 17 mm longer than males, whereas the biggest difference in SCL was observed in Batna (23 mm). The greatest degree of SSD was presented by SH, where female shells were 21–26% higher than those of conspecific males (Appendix A in Appendix A). 

#### 3.1.2. Sexual Shell Shape Dimorphism (SShD)

Multivariate comparisons using the size-corrected variables with SCL as a covariate revealed overall shape variations between sexes without reference to their geography (MANCOVA, sex Wilk’s *λ* = 0.44, *F*_7, 369_ = 67.54, *p* < 0.001). The covariates analysis for corrected body sizes (ANCOVAs, with SCL as a covariate) showed that six out of seven examined shell characteristics were significantly different between males and females independent of their geographic origin (Table 2). Reflecting a difference in internal volume, females exhibited wider, more voluminous, and more domed shells compared to males (Table A2 in Appendix A). The only exception was the curvilinear length of the carapace (CCL, expressed relative to SCL), which was not sexually dimorphic. The curved carapace length resulted from both the domed carapace and recurved supracaudal scute, presumably stretched in an opposite direction for each of the two sexes. Indeed, females presented a greater SH than males, which, in contrast, displayed a longer recurved supracaudal scute, leading to the invariability of CCL characteristics.

Additionally, SShD was significant in all localities under MANCOVA using morphometric size-corrected variables, with SCL as a covariate (MANCOVA, sex × site Wilk’s *λ* = 0.12, *F*_77, 2158_ = 11.90, *p* < 0.001). Comparisons by characteristics using two-way ANCOVA (sex × site) were significant (Table 2), but only modestly so, not exceeding 10% in all general shell shape characteristics (Table A2). The most sexually dimorphic general characteristic of shell shape in all six populations was the plastral cover of the carapace length (corrected PL), which was larger in females. A shorter plastron indicates that males have developed mathematically larger openings for the movement of the head and tail. The specific characteristic of ANW, used to estimate the space available for tail movement, was larger in males (endowed with long tails) than in females in all studied populations, differing by 20–28% (Table A2 in Appendix A). 

### 3.2. Geographic Variation in Body Size and Shell Shape

#### 3.2.1. Geographic Variation in Body Size (GVS) 

When analyzed separately, both sexes differed significantly between the studied localities in all measured body size characteristics (MANOVA male, site Wilk’s *λ* = 0.18, *F*_40, 682_ = 8.08, *p* < 0.001; female, site Wilk’s *λ* = 0.23, *F*_40, 857_ = 8.54, *p* < 0.001). The results of the analysis by characteristics (ANOVAs) were similarly significant and are recapitulated in Table 2. The Tukey post hoc test (comparison of means between each pairwise combination of groups) for the straight carapace length between the localities revealed that only the Djelfa population was significantly different from all other populations, separating Djelfa into one group and the rest of the localities into another (Table 3).

Degrees of GVS for all measured general descriptors were 16–21% and 14–19% for males and females, respectively (Table A3 in Appendix A). Both males and females from Djelfa had significantly longer carapaces than specimens from other populations and were 16.0% and 15.9% longer than males and females, respectively, from the population with the smallest tortoises (based on mean SCL values). When we excluded data from Djelfa, neither males nor females differed in SCL between localities (ANOVA male, site: *F*_(4, 153)_ = 0.38, *p* = 0.82; ANOVA female, site: *F*_(4, 127)_ = 0.62, *p* = 0.65). This reinforces the exceptionality of tortoise morphology in the Djelfa locality (see also Figure 3 and Table 3), as all body size characteristics were greater in the Djelfa population, with the tortoises there displaying longer, wider, and higher shells than those in all other studied populations (Table A1 in Appendix A). The largest male and female had an SCL of 186 and 230 mm, respectively (Table A4 in Appendix A).

#### 3.2.2. Geographic Variation in Shell Shape (GVSSh) 

After size correction, the MANCOVA comparisons (with SCL as a covariate) revealed an overall shell shape variation between localities in each of the two sexes (MANCOVA male, site Wilk’s *λ* = 0.24, *F*_35, 658_ = 7.59, *p* < 0.001; female, site Wilk’s *λ* = 0.28, *F*_35, 826_ = 8.27, *p* < 0.001). Details of the one-way ANCOVA comparisons are recapitulated in Table 2. Similar to GVS, the majority of shell shape analyses showed convergent trends across populations. Djelfa tortoise bodies were higher and more domed than those from populations in other sites; the males and females in Djelfa were 16–22% and 15–19% larger, respectively, than those from the smallest population (Table A3 in Appendix A).

### 3.3. Timing of Attainment of Maturity and Its Implication on Adult Body Size Variations

Sexual maturity, estimated by counting growth rings, was always reached earlier in males than in females (Table 4), with a mean time difference of at least 1.4 years recorded in Djelfa, up to a maximum of 3.0 years at Laghouat. Remarkably, Djelfa tortoises presented the latest maturity time since males became mature at 9.1 years (range, 8–11 years) and females at 10.5 years (range, 9–12 years). The remaining five populations had comparable sexual maturity times. The earliest time for attainment of maturity was observed in Batna, where males matured at 6.3 years (range, 6–7 years) and females at 8.4 years (range, 8–9 years). 

### 3.4. Demographic Mechanisms (Proximal) Influencing Adult Body Sizes

#### 3.4.1. Determinism of Body Size Acquired up to the Onset of Maturity

The mean values of body size for both males and females, calculated among the individuals that had just reached sexual maturity (SCLα) and at adult age (SCL), were significantly related (male: r *Pearson* = 0.94, *p* < 0.01, *n* = 6; female r *Pearson* = 0.90, *p* < 0.05, *n* = 6) (Figure 4). When data for all localities were grouped together, body size at the onset of maturity differed between sex (ANOVA: *F*_(1, 67)_ = 9.08, *p* < 0.01). Females with a comparably longer time to maturity also reached sexual maturity at a larger size (range of means, 117.1–143.9 mm) than males (range of means, 113.3–131.5 mm) in all sampled populations (Table 4). The mean adult body size (dependent variable) according to the corresponding SCLα (covariate) was also significantly different between the sexes (independent variables) (ANCOVA: *F*_(1, 9)_ = 34.26, *p* < 0.001). Therefore, statistical tests validated the implication of size at maturity, which is dependent on the duration of growth up to the onset of maturity in sexual variations of adult body sizes. Compared to smaller male body sizes, females benefited from a longer growth period before reaching maturity and, consequently, had larger body sizes at both maturity and adulthood.

Due to a significantly longer juvenile growth period (see above), tortoises from the Djelfa locality acquired a longer SCLα than their conspecifics from the remaining localities, with males and females up 16% and 23% longer, respectively, than the smallest population (calculated from the SCLα data in Table 4). The mean body sizes at maturity showed geographical variation between males (ANOVA: *F*_(5, 28)_ = 2.85, *p* < 0.05) and females (ANOVA: *F*_(5, 29)_ = 7.57, *p* < 0.001), and as shown above, were significantly related to adult sizes. Thus, the differences in body sizes at maturity could explain the geographic variations in adult tortoise bodies. 

#### 3.4.2. Determinism of Body Size Acquired at Adulthood 

Two phases of growth clearly demonstrated rapid juvenile growth up to the onset of maturity, followed by continuous, slow adult growth (Table 4). The deduced mean adult growth was sexually different (ANOVA: *F*_(1, 10)_ = 38.43, *p* < 0.001); post-maturity females continued to grow slowly in size but with 30–90% greater growth on average compared to males (Table 4). However, the acquired growth of tortoises during this period was not responsible for the corresponding adult SSD. The ANCOVA (sex as the independent variable) of mean SCL (dependent variable) according to the acquired growth at adulthood (covariate) was not statistically different (*F*_(1, 9)_ = 2.32, *p =* 0.162). The deduced adult growth was also not responsible for the observed geographic variation in adult sizes (male, r *Pearson* = 0.46, *p* = 0.362, *n* = 6; female, r *Pearson* = −0.02, *p* = 0.967, *n* = 6) (Figure 4).

### 3.5. Ecological Correlates of Morphometric and Demographic Variations 

An exploratory examination using principal components analysis was conducted to visualize, in a fairly straightforward manner, the clustering trends obtained among the different populations with regard to the contributing ecological variables of body size and shape, as well as age of maturity. Figure 5 provides a multidimensional environmental description of these sizes and ages, based on the following most contributory variables: altitude, Eff-T_F_, PCI-D, PCI-W, and T-Max March. 

However, neither body size nor age at maturity measured in the different populations was significantly related to the five primary climatic variables tested between each pairwise combination of groups. No significant correlation was found (*p* ≥ 0.05) between any of the observed morphometric and demographic variations of tortoises (body size variables, shell shape variables, SSD%, SShD%, age at maturity, SCL at maturity, SDSα%, SDTα, acquired growth at adult-age, and GVS%) and geographic and climate variables (latitude, longitude, elevation, temperature indexes, precipitation indexes, and A_dM_).

## 4. Discussion

Remarkable variability exists in body size and shell shape in *T. graeca* through its distribution range, e.g., [7,8,10,34,35]. Our study, based on a large sample, supports this observation, with the Djelfa population containing tortoises displaying the largest shells, close to record sizes known for Western Mediterranean *T. graeca* [12,36]. Size at maturity depends on the duration of growth from hatching until reaching maturity, which differs between both sexes. Nevertheless, we found no relationships between adult body sizes of Algerian *T. graeca* and the tested geographic and climatic variables.

### 4.1. Sexual Size and Shell Shape Dimorphism

Size in the studied *T. graeca* population was biased towards females, which were found to be longer, wider, and also higher than males, corresponding with the usual traits in the genus *Testudo* [8,9,13,18,37], for which the only exception is *T. marginata,* where no sex predominates [13]. *T. hermanni*, *T. horsfieldii*, and *T. kleinmanni* differ by having a posterior carapace width larger in males [13,18,37]. The wide rear half of the body in these three species is thought to provide great stability and power in male-to-male combat. These fights can last more than one hour [23]. In comparison, males of *T. graeca* have less aggressive courtship behavior, with shorter male-to-male combat [8,13]. This behavioral difference may cause the different patterns observed in posterior body width. 

Similar to all other *Testudo* species, ANW was larger in males, e.g., [8,18,23,37], making this trait useful in the sexing of adults [22]. This space allows males unrestricted tail movement in order to reach the female cloaca orifice, whereas females merely need a space allowing the passage of eggs [13]. 

Natural selection for greater fecundity supports larger females having greater biotic capacity; a larger body accommodates larger clutches, egg sizes, or annual egg production [21,38]. The clutch size and body size of *T. graeca* females were positively correlated [11]. Sexual selection for courtship behavior can result in both large or small male sizes. Male reproductive success increases with their ability to fertilize females, often deserved as the result of body-strength combat between rivals, resulting in the selectivity of large male bodies [21]. In the same tortoise species, a small body size may help males increase the efficiency of their locomotion, which may allow them to move faster and thus facilitate greater agility in searching for and engaging with females for copulation [13,39]. 

Females have larger, more domed, wider, and more closed shells than males [7,8]. These characteristics gives females more volume for the visceral organs involved in the nutrient treatment necessary for vitellogenesis, gestation, and carrying eggs [18]. A strong positive correlation was observed between the fecundity of females and their shell shape [18,38]. Males are not constrained by any parental care but concerned with courting and mating. Shorter male plastrons relative to carapace length indicate larger openings for head and tail movement, which are, respectively, necessary to increase the ability of males to right themselves when overturned during fighting and achieve success in mating [7,13]. Indeed, males were able to right themselves more quickly than females in experimental tests [18]. Additionally, a larger ANW relative to carapace length allows males more space for ventral and lateral movements of the tail and, consequently, success in depositing sperm in the female cloacae [7,13]. A balance between natural and sexual selection may explain the invariability in the curvilinearity of the carapace between the two sexes, where females have a domed carapace and males have a more recurved supracaudal scute to enhance inward tail movement [7,13]. Thus, sexual differences in adult shell shape are not just allometric effects but reflect adaptive responses to different selective contexts.

SSD was also found to be more pronounced than SShD in some previously studied populations of *T. graeca*, *T. hermanni*, and *T. horsfieldii* [7,8,13,18,19]. The SSD range that we observed (13–26%, based on SCL) is comparable with the only available Algerian study, Tunisian, Moroccan, and Spanish populations, whose SSD ranges between 9 and 21% [8,9,15,40,41]. 

*Testudo graeca* females in the present study reach sexual maturity later than males, similar to the El Kala population, Northern Algeria [15], as well as Moroccan and Spanish populations [6,12]. According to the latest studies, sexual maturity was reached at 5–9 years of age in males and 6–14 years in females. On average, more than one additional year of growth allows females to reach a larger adult size than males. Females need to reach a greater size to increase their biotic capacity in readiness for their first reproduction. The decline in growth rates after reaching maturity indicates a reorientation of energy allocations from growth to reproduction [6,23]. 

### 4.2. Geographic Variation in Body Size and Shell Shape

The morphometry of *Testudo* tortoises often varies between populations [14,19,42], which is also true for *T. graeca* [8,9]. The variability of adult body size was calculated to be 16.0% for both males and females among Algerian populations as a mean factor of carapace length difference between populations of the largest and smallest body size. Higher values of SCL variability were observed in Moroccan populations, with a 20.4% difference in males and a 20.0% difference in females [8].

Tortoises from Djelfa are the second largest among *T. graeca* conspecifics in the entire Western Mediterranean; the largest male and female measured SCL values were 186 and 230 mm, respectively. The record-sized specimens in this area were recorded in a small population from Tagourast (Boulemane Province) in Southeastern Morocco, containing a male with an SCL of 200 mm and two females of 236 and 249 mm [36], whereas the record-sized male originated from Spain, where the SCL of the longest male and female measured 191 and 211 mm, respectively [12]. The largest tortoises from Spain presented later attainment of maturity, with a mean age of maturity of 6.9 years (a range of 5–9) and 8.5 years (a range of 6–14) for males and females, respectively. However, the picture was different among Moroccan populations, where the largest tortoises in the Admine and Essaouira populations reached maturity more than 2 years earlier than those in the Jbilet population, which is composed of medium-sized adults that reach maturity at 7.6 years (6–10.2) for males and at 10.5 years (8–14) for females [9].

The late maturations recorded in Djelfa tortoises for males at 9.1 years (8–11) and females at 10.5 years (9–12), respectively, were determinants for the exceptional sizes observed in adulthood. Similar correlations were also found between *T. horsfieldii* and *T. hermanni* but without a direct significant relationship between body size at maturity and in adulthood [14,23]. Willemsen and Hailey [14] proposed that individual variations in age and size at the attainment of maturity reflect the phenotypic plasticity of individual growth and maturation. The plasticity of growth and maturation gives chelonians different versions of SSD and SShD depending on the extent of environmental pressure to which they are exposed. The most involved biotic factors are variations in trophic resources [24], competition, and predation [43]. We did not measure the impact of these factors on *T. graeca* populations. However, the lack of a tree stratum or dense bush in pre-desert areas (Batna, Aflou, and Laghouat) or open maquis (Annaba and El Kala) suggests that the pressure of the three factors may be more prominent in such areas compared to that in the Djelfa locality, which may offer more shelter and resources. The over 190 km^2^ Senelba forest at Djelfa is the last bastion of trees before the onset of the desert and is protected by foresters against habitat fragmentation and illegal practices. The structural complexity of vegetation, from areas with sparse vegetation to denser and compact forests [44], affords tortoises exposure to the sun and shelter from the sun in shadier places, as needed. 

Despite some previous authors explaining tortoise plasticity as the result of abiotic factors, especially climatic effects [8,45], the adult body size variability of Algerian *T. graeca* has no correlation to geographic and climate variables. Carretero et al. [8] assumed that the body size variations recorded among Moroccan populations differed depending on annual rainfall and thermal amplitude rather than the latitudinal gradient. Willemsen and Hailey [14] proved that the body size of Greek *T. hermanni* increases the further North and the higher the altitude they are found, with an inverse relationship to temperature, whereas body size has no significant correlation with longitude or rainfall. Conversely, *T. hermanni* from South Serbian populations were larger than tortoises from the North [42]. All these contradictory conclusions contribute to an absence of general rules explaining the implications of ecological conditions on the geographical variability of tortoise morphology.

## 5. Conclusions

The body sizes and shell shapes of Algerian *T. graeca*, inhabiting a range of environments, display different morphologies. We found clear evidence that body size acquired up to the onset of maturity determines the obtained size in adulthood. Size at maturity depends on the duration of growth and seems to be involved in the sexual dimorphism of adult body sizes. Due to a significantly longer juvenile growth period, tortoises from the Djelfa locality acquired larger bodies than their conspecifics from the other studied localities. An analysis of feeding resources and a follow-up of tortoise activity in this population could bring more certainty to our findings. Moreover, the phylogeographic situation of the studied populations, especially in the South, is uncertain compared to Northern populations. However, the strong morphological differences in the Djelfa population composed of larger individuals, which suggest a clear adaptation to the semi-arid climate and benefits from a high level of ecological balance, cast doubts on the taxonomic implications of this population. Until information on the Djelfa population and other populations inhabiting different regions in Algeria becomes available, especially in the Southwest, these populations should be treated as urgent conservation units regardless of their taxonomic status.

## Figures and Tables

**Figure 1 animals-12-01330-f001:**
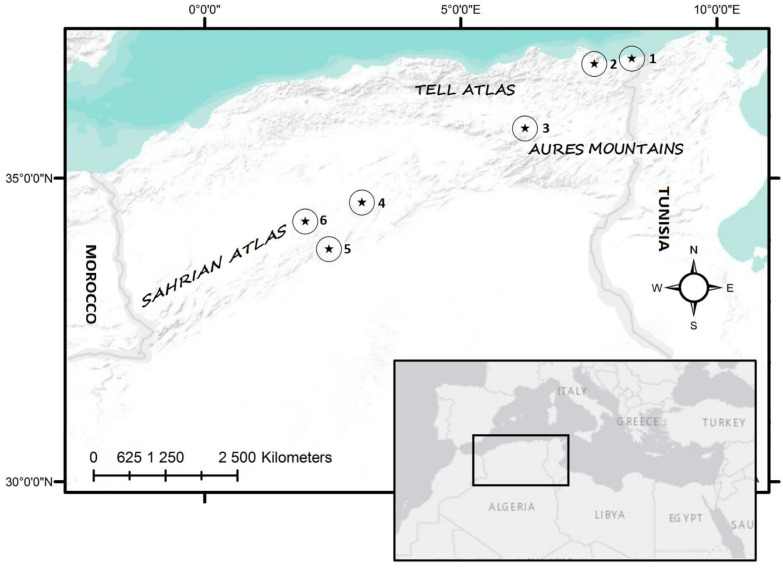
Placement of the studied localities: 1–El Kala (Boumalek); 2–Annaba (Hadjar Eddis); 3–Batna (Ain Yagout); 4–Djelfa (Senalba Chergui); 5–Laghouat (Tadjmout); 6–Aflou (Gueltet Sidi Saad).

**Figure 2 animals-12-01330-f002:**
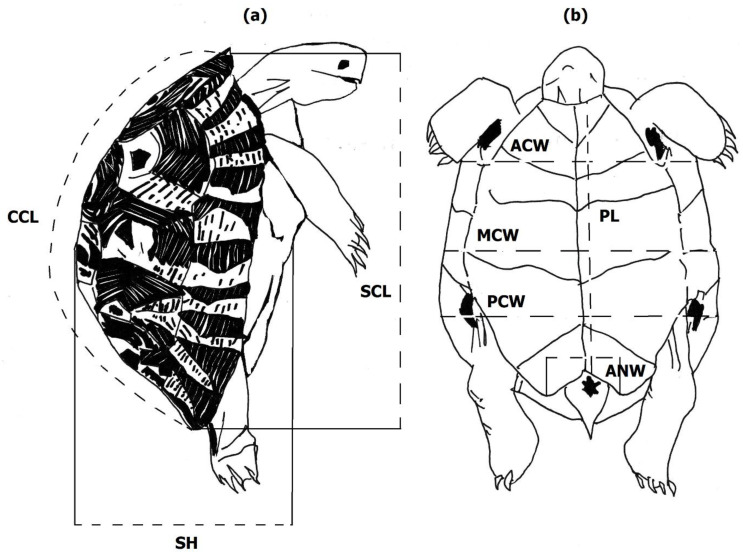
Selected body size descriptors: (**a**) lateral view; (**b**) ventral view. Straight carapace length (SCL; the straight anteroposterior distance between the nuchal and supracaudal scutes); curved carapace length (CCL; the midline distance along the curved carapace from the nuchal to the tip of the supracaudal scutes); mid-body carapace width (MCW; body width in the middle of the abdominal Scheme 7. th marginal scute); anterior carapace width (ACW; shell width at the junction of the gular and humeral scutes in the midline); posterior carapace width (PCW; shell width at the junction of the femoral and anal scutes in the midline-usually the maximum width); midline plastron length (PL; the straight anteroposterior distance from the front of the gular scutes to the back of the anal scute); shell height (SH; the maximum vertical measure from the top of carapace to plastron); and anal notch width (ANW; the width between the tips of the 2 anal scutes).

**Figure 3 animals-12-01330-f003:**
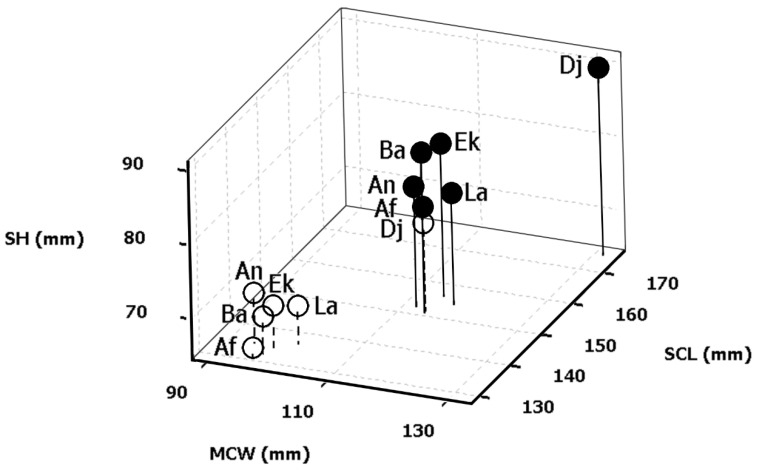
Three-dimensional scatterplot of body size distances between localities and sexes of Algerian *T. graeca*. Black circles with continuous sticks are means of females. White circles with discontinuous sticks are means of males. SCL—straight carapace length; MCW—mid-body carapace width; SH—shell height; Ek—El Kala; An—Annaba; Ba—Batna; Dj—Djelfa; La—Laghouat; Af—Aflou.

**Figure 4 animals-12-01330-f004:**
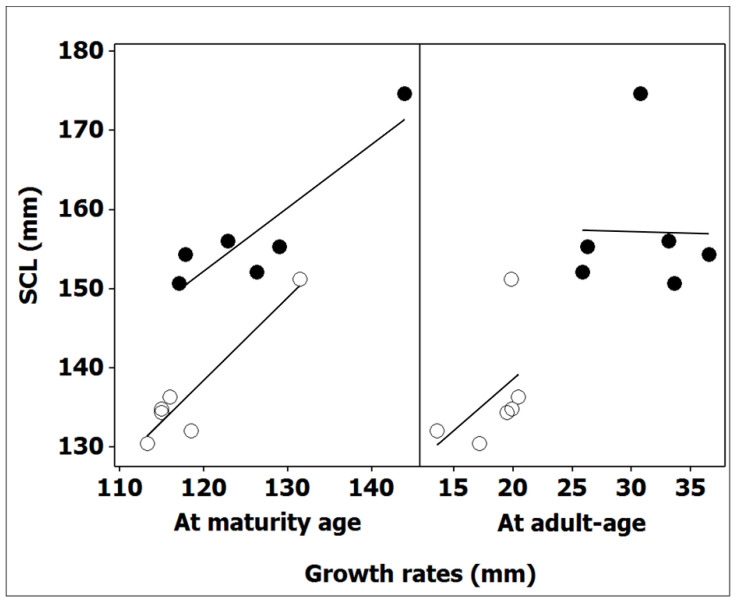
Relationship between body size in adult males (white circles) and females (black circles) and growth rates acquired at both the time of sexual maturity and during the adult life of Algerian *T. graeca.* Equations of regression lines are for males and females: Y1 = 12.8 + 1.05X (r *Pearson* = 0.94, *p* < 0.01, *n* = 6), Y2 = 113.0 + 1.31X (r *Pearson* = 0.46, *p* = 0.362, n = 6) and Y1 = 55.7 + 0.80X (r *Pearson* = 0.90, *p* < 0.05, *n* = 6), Y2 = 159.0 − 0.05X (r *Pearson* = −0.02, *p* = 0.967, *n* = 6), respectively. Significant correlations are highlighted by underlining.

**Figure 5 animals-12-01330-f005:**
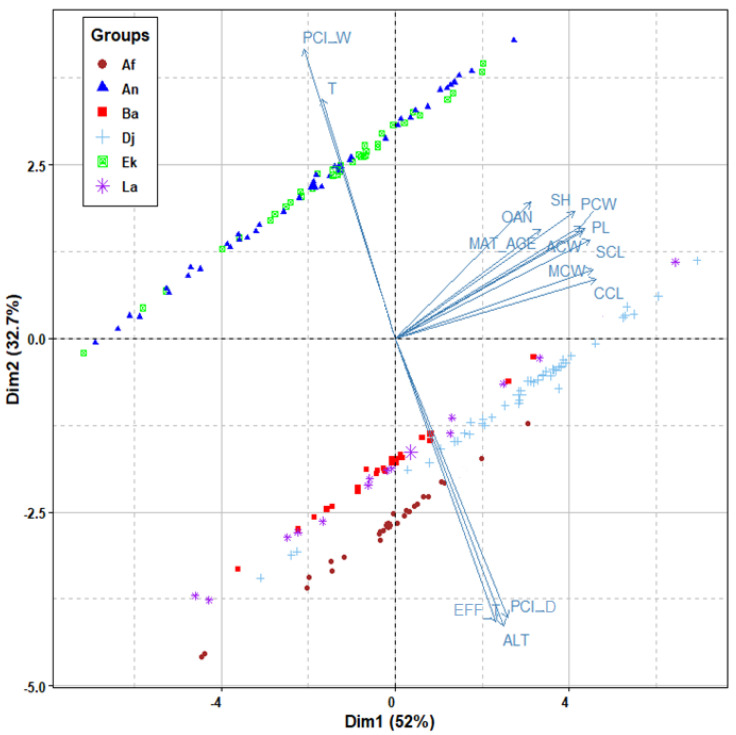
PCA of the tortoise body sizes of adult females and their age at maturity in the six studied populations in relation to the most contributory climatic variables. Alt: altitude, EFF-T: effective temperature, PCI D: precipitation concentration index for the dry season, PCI W: precipitation concentration index for the wet season, T: mean daily maximum temperature for March, Ek: El Kala; An: Annaba; Ba: Batna; Dj: Djelfa; La: Laghouat; Af: Aflou. Abbreviated characteristics of body size are explained in the Materials and Methods section.

**Table 1 animals-12-01330-t001:** Characterization of studied localities.

Study Region ‘Specific Study Site’	Altitude	Localisation	Regional Climate	Dominant Habitat	Vegetation	Anthropogenic Influence
**El Kala** ‘Boumalek’	12–15 m	Extreme Northeastern Algeria	Sub-humid	Low shrubland on sandy soil	Mediterranean, grass and low shrubland	- Partially inhabited site - Important farming
**Annaba** ‘Hadjar Eddis’	45–90 m	Northeastern Algeria	Sub-humid	Low shrubland on siliceous soil	Mediterranean, grass and low shrubland	- Near agglomeration - No recent farming
**Batna** ‘Ain Yagout’	820–950 m	Central Aures Montain	Cool inferior semi arid	Degraded steppe	*Stipa tenacissima, Artemisia herba-alba,* and *Lygeum spartum* plant formations	- Overgrazing - Far from agglomeration
**Djelfa** ‘Senalba Chergui’	1240–1300 m	Central Saharan Atlas	Cool inferior semi arid	Natural pine forest	Three vertical layers-trees, shrubs, and grass with herbaceous plants	- No farming or settlements - Rigorous legal protection - Pasture regulated
**Laghouat** ‘Tadjmout’	900–950 m	Southern foothills of the central Saharan Atlas	Cold inferior arid	Pre-desert steppe	Low vegetation cover of lower strata	- Overgrazing combined with prolonged droughts - Sedentary livestock breeding
**Aflou** ‘Gueltet Sidi Saad’	1100–1200 m	Southwestern Saharan Atlas	Cool inferior semi arid	Degraded steppe	Typical shrub and herbaceous plants of steppe	- Overgrazing - Far from agglomeration

**Table 2 animals-12-01330-t002:** Sexual and geographic differences in body size and shell shape among adults of Algerian *T. graeca*. *F*_(*m*−1, *e*)_ and *p* values for body size are from ANOVA, and for shell shape are from ANCOVA (*e*, error degrees of freedom; Sample size (*n*) = *e* + *m* (number of groups being compared) + 1 (if the ANCOVA test case); ^a^ = One-way interaction; ^b^ = Two-way interaction; * = *p* < 0.05; ** = *p* < 0.01; *** = *p* < 0.001).

	ANOVA for Body Size								ANCOVA for Shell Shape (SCL as Covariate)							
	Sex ^a^		Sex × Site ^b^		Male, Site ^a^		Female, Site ^a^		Sex ^a^		Sex × Site ^b^		Male, Site ^a^		Female, Site ^a^	
	*F* _1, 379_	*p*	*F* _11, 369_	*p*	*F* _5, 164_	*p*	*F* _5, 205_	*p*	*F* _1, 378_	*p*	*F* _11, 368_	*p*	*F* _5,163_	*p*	*F* _5, 204_	*p*
SCL	111.5	***	20.9	***	9.6	***	8.7	***								
CCL	102.7	***	24.2	***	13.2	***	12.2	***	1.0	0.32	12.3	***	18.5	***	9.9	***
PL	209.4	***	31.2	***	11.1	***	7.1	***	100.8	***	11.3	***	3.4	**	1.4	0.22
SH	183.0	***	26.8	***	6.8	***	7.8	***	57.4	***	8.4	***	1.7	0.14	4.9	***
MCW	148.0	***	34.7	***	17.2	***	15.9	***	28.2	***	10.9	***	7.5	***	9.0	***
ACW	145.1	***	29.8	***	12.2	***	13.3	***	26.2	***	7.3	***	2.9	*	6.7	***
PCW	146.0	***	26.7	***	11.4	***	9.9	***	26.7	***	5.8	***	2.2	0.06	4.5	**
ANW	98.3	***	11.1	***	3.0	*	2.9	*	334.8	***	40.3	***	6.6	***	6.1	***

**Table 3 animals-12-01330-t003:** Tukey Post hoc test (HSD) of straight carapace length (SCL) between localities at 95% confidence interval. Ek—El Kala; An—Annaba; Ba—Batna; Dj—Djelfa; La—Laghouat; Af—Aflou. Signification: * = *p* < 0.05, ** = *p* < 0.01, *** = *p* < 0.001.

Contrast	Diff between Means (mm)	Standardized Difference	Pr > Diff	Signification	Lower Bound (95%)	Upper Bound (95%)	Lower Bound (95%)	Upper Bound (95%)
Dj vs. Af	23.83	6.39	**<0.0001**	***	13	35		||||||||||||||||||||||| ||||||||||||||||||||||||||||||||||||
Dj vs. Ba	22.65	5.95	**<0.0001**	***	12	34		||||||||||||||||||||| |||||||||||||||||||||||||||||||||||||
Dj vs. Ek	19.02	5.26	**<0.0001**	***	9	29		||||||||||||||||| |||||||||||||||||||||||||||||||||
Dj vs. La	18.20	3.37	**0.011**	*	3	34		||||| |||||||||||||||||||||||||||||||||||||||||||||||||||||
Dj vs. An	18.11	4.63	**<0.0001**	***	7	29		|||||||||||||| ||||||||||||||||||||||||||||||||||||
An vs. Af	5.72	1.38	0.742	No	−6	18	|||||||||||	||||||||||||||||||||||||||||||
An vs. Ba	4.53	1.08	0.888	No	−7	17	|||||||||||||	||||||||||||||||||||||||||||
An vs. Ek	0.90	0.22	1.000	No	−11	13	|||||||||||||||||||	||||||||||||||||||||||
An vs. La	0.08	0.01	1.000	No	−16	16	||||||||||||||||||||||||||||	||||||||||||||||||||||||||||
La vs. Af	5.64	1.01	0.915	No	−10	22	||||||||||||||||||	|||||||||||||||||||||||||||||||||||||
La vs. Ba	4.45	0.79	0.969	No	−12	21	||||||||||||||||||||	|||||||||||||||||||||||||||||||||||
La vs. Ek	0.82	0.15	1.000	No	−15	17	||||||||||||||||||||||||||	|||||||||||||||||||||||||||||
Ek vs. Af	4.82	1.24	0.814	No	−6	16	|||||||||||	|||||||||||||||||||||||||||
Ek vs. Ba	3.63	0.93	0.939	No	−8	15	|||||||||||||	|||||||||||||||||||||||||
Ba vs. Af	1.19	0.29	1.000	No	−10	13	||||||||||||||||||	||||||||||||||||||||||

**Table 4 animals-12-01330-t004:** Age and body size at maturity and growth during adulthood differed between the sexes of Algerian *T. graeca*. Here, the results for age (year) and body size (millimeter) are taken as means, and those in parenthesis are ranges. *F* and *p* values are from a one-way ANOVA (*e*, error degrees of freedom; Sample size (*n*) = *e* + 2. *** = *p* < 0.001; SDTα, Sexual difference in time of maturity is calculated here as (Female age at maturity − Male age at maturity); SCLα, straight carapace length at maturity; SDSα%, sexual size dimorphism at maturity is calculated as 100 ((Female SCLα − Male SCLα)/Male SCLα); SDG%, sexual difference in growth to adulthood after maturity is calculated as 100 ((Female growth − Male growth)/Male growth).

	Age at Maturity						Body Size (SCL) at Maturity			Growth during Adult Phase		
	Male	Female	*F*	*p*	*e*	SDTα	Male SCLα	Female SCLα	SDSα%	Male	Female	SDG%
Ek	6.9 ± 0.9 (6–9)	9.5 ± 0.9 (8–11)	148.4	***	67	2.6	115.0	122.9	6.9	19.4	33.1	70.7
An	6.7 ± 1.0 (6–9)	9.0 ± 0.8 (8–10)	96.9	***	60	2.3	114.9	126.3	9.9	19.9	25.8	30.0
Ba	6.3 ± 0.5 (6–7)	8.4 ± 0.5 (8–9)	259.5	***	51	2.1	118.5	129.0	8.9	13.5	26.3	94.4
Dj	9.1 ± 0.8 (8–11)	10.5 ± 1.0 (9–12)	44.9	***	80	1.4	131.5	143.9	9.5	19.8	30.8	55.6
La	6.4 ± 0.7 (6–8)	9.4 ± 1.0 (8–11)	75.5	***	26	3.0	115.9	117.8	1.6	20.4	36.5	79.2
Af	6.3 ± 0.5 (6–7)	8.4 ± 0.5 (8–9)	247.0	***	51	2.1	113.3	117.1	3.3	17.1	33.6	96.7

## Data Availability

The data presented in this study are available on request from the corresponding author. The data are not publicly available because the authors will use them in future research.

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
