# Peer review of "Mechanisms Determining Body Size and Shape Difference in Algerian Spur-Thighed Tortoises (Testudo graeca)"

_animals, 2022, doi:10.3390/ani12101330_

Round 1

Reviewer 1 Report

The significant improvement of the manuscript has been done with introducing info on genetic structuring of this species. But, something else is questionable: in the Intro part of the ms, an interesting reference about T. graeca from other part of Algeria was mentioned  - it is population study, and genetic background of that population is known (reference no. 15). That population has been analysed for its structure and demography and in this study the authors also presented some demographic data of analysed populations. Why they did not compare values of demographic parameters common for both studies and discuss the outcomes? My opinion is that this reference (under the no. 15) is not something that just has to be mentioned in the Introduction, but that it has to be discussed for the common parameters and its values.

Author Response

The significant improvement of the manuscript has been done with introducing info on genetic structuring of this species. But, something else is questionable: in the Intro part of the ms, an interesting reference about T. graeca from other part of Algeria was mentioned  - it is population study, and genetic background of that population is known (reference no. 15). That population has been analysed for its structure and demography and in this study the authors also presented some demographic data of analysed populations. Why they did not compare values of demographic parameters common for both studies and discuss the outcomes? My opinion is that this reference (under the no. 15) is not something that just has to be mentioned in the Introduction, but that it has to be discussed for the common parameters and its values.

Reply: Results of the indicated reference (Rouag et al. 2007) in relation to our study have been compared and discussed between lines 494 and 501: “The SSD range that we observed (13-26 %, based on SCL) is comparable with the only available Algerian study, Tunisian, Moroccan, and Spanish populations, whose SSD ranges between 9 and 21% [8, 9, 15, 40, 41].

Testudo graeca females in the present study reach sexual maturity later than males, similar to El Kala population, Northern Algeria [15], as well as Moroccan and Spanish populations [6, 12]. According to this latest studies, sexual maturity was reached at 5-9 years of age in males, and 6-14 years in females.

Reviewer 2 Report

Thanks to the authors for their convincing replies to all my comments and remarks about all details related to both form and content of the Ms.

Author Response

Thank to reviewer.

This manuscript is a resubmission of an earlier submission. The following is a list of the peer review reports and author responses from that submission.

Round 1

Reviewer 1 Report

This is valuable piece of scientific work, especially as there were no such detailed analysis performed on T. graeca in Algeria. however, I would have two comments which would help in the design of the future studies:

1) for the purposes of comparison of  of populations in different environments, at least two populations from the same environment should be analysed and at least two different environments should be covered by the study-  to distinguish between intra-group variation and inter-group variation. Of course, when analysing populations in the field, it is sometimes difficult to apply such a design. Here, you could additionally compare El Kala and Annaba (group 1) with Batna and Aflou (group 2), to check for intragroup and intergroup differences.

2) My second comment would be to include the slope of the terrain into the analysis, combined with the habitat type defined simply as "open" (meadow, steppe) or "closed" (forest or dense shrubs). It would be interesting for comparison of interpopulation differences regarding degree of SShD.

I recommend you to read also this article:

https://brill.com/view/journals/ab/63/4/article-p381_1.xml

because it also discusses differences in shell size and shape in the context of functionality and different habitat types. As there requirements for the study design recommended to you were not fulfilled, here you have opportunity with your data to check proposed hypotheses and thus to increase quality of your paper. Your study is very important for future conservation planning in the area.

Reviewer 2 Report

see the comments in the attached file.

Reviewer 3 Report

Review – Tiar Saadi et al

This is an interesting data set that use annuli counts and morphometric data  from spur-thighed tortoises to evaluate, sexual size-dimorphism, age, and size of reproduction in several different populations across an environmental gradient. The authors suggest that there is a gradient effect and that natural selection has contributed to the differences reported among these populations. I have several points that I think the authors should consider in a substantial revision of this manuscript.

First, the authors claim there is a gradient effect but there is really one population that is different from the rest that is driving the entire analysis. When the Djelfa population is removed from the analysis not effects were differences among the remain populations was detected Line 293. This suggests that there is something unique about the Djelfa population and not really a gradient. This is further corroborated by the fact that there were no correlations between any of the environmental variables with the observed differences among the populations. I might suggest that there is something unique about the resource environment at Djelfa that results in the differences between the populations. The suggestion that this is a more lush forest habitat vs desert scrub of the other sites attests to this observation.

Second, a multivariate approach would really be more appropriate for this data set. There is high correlation structure within the morphometric data set that warrants a multivariate approach. I applaud that the authors use an ANCOVA to adjust for body size, that is the right tool, but should be a MANCOVA. Also, in the presentation of the MANCOVA results please make sure to present the results of the homogeneity of variance test as this is a critical prerequisite to using and interpreting the ANCOVA. Also, some sort of post-hoc test to evaluate which populations are different from one another would be highly appropriate. It may further support that only the Djelfa population is different. Additionally, a multivariate or perhaps Bayesian model fitting analysis may be more appropriate when evaluating the influence of environmental variation on population differences.

Third, the benchmark used to determine age at maturity is not clearly identified other than to say the distance between annuli “become narrower” line 160. Can you provide a more quantitative criteria to determine age at maturity. Furthermore, many species of turtles can have both appreciable growth and reproduce. This assumption should be explicitly stated and how this possibility was handled in the data set.

Lastly, the paper needs considerable work in grammar, sentence structure, and proof reading for legibility. I found I was often confused by the way in which the authors presented certain information.

My suggestion is that the authors do the appropriate multivariate analyses and then resubmit the paper. The current environmental analysis is flawed (700 correlations) and requires a more appropriate investigation of the data.

More specific Comments:

Abstract – “delayed age at maturity was associated with larger body size – cannot say that the delayed maturity caused large body size – no data to support this statement.

Line 59 – use higher fecundity

Line 65-67 – very awkward statement that reads like a tautology

Table !. Suggest removing the latitude and longitude data to protect these populations from poachers. The map should be good enough.

Authors present metrics of SSD% ans SShD% but do not identify in the methods how these values are calculated.

I do not understand Figure 4, particularly the left-hand panel

Line 167 – why is there a reference to Bayesian inference here – none is presented in this paper